# Novel Organic–Inorganic Nanocomposite Hybrids Based on Bioactive Glass Nanoparticles and Their Enhanced Osteoinductive Properties

**DOI:** 10.3390/biom14040482

**Published:** 2024-04-16

**Authors:** Nicolás Cohn, Henrik Bradtmüller, Edgar Zanotto, Alfredo von Marttens, Cristian Covarrubias

**Affiliations:** 1Departamento de Química, Facultad de Ciencias Naturales, Matemáticas y del Medio Ambiente, Universidad Tecnológica Metropolitana, Santiago 7800003, Chile; 2Center of Research, Technology and Education in Vitreous Materials, Department of Materials Engineering, Federal University of Sao Carlos, Sao Carlos 13565-905, SP, Brazil; mail@bradtmueller.net (H.B.); dedz@ufscar.br (E.Z.); 3Oral and Maxillofacial Implantology Program, Graduate School, Faculty of Dentistry, University of Chile, Santiago 7520355, Chile; 4Laboratory of Nanobiomaterials, Institute for Research in Dental Sciences, Faculty of Dentistry, University of Chile, Santiago 8380544, Chile

**Keywords:** hybrid biomaterial, bioactive glass nanoparticles, nanocomposite scaffolds, bone regeneration

## Abstract

Inorganic–organic hybrid biomaterials have been proposed for bone tissue repair, with improved mechanical flexibility compared with scaffolds fabricated from bioceramics. However, obtaining hybrids with osteoinductive properties equivalent to those of bioceramics is still a challenge. In this work, we present for the first time the synthesis of a class II hybrid modified with bioactive glass nanoparticles (nBGs) with osteoinductive properties. The nanocomposite hybrids were produced by incorporating nBGs in situ into a polytetrahydrofuran (PTHF) and silica (SiO_2_) hybrid synthesis mixture using a combined sol–gel and cationic polymerization method. nBGs ~80 nm in size were synthesized using the sol–gel technique. The structure, composition, morphology, and mechanical properties of the resulting materials were characterized using ATR-FTIR, ^29^Si MAS NMR, SEM-EDX, AFM, TGA, DSC, mechanical, and DMA testing. The in vitro bioactivity and degradability of the hybrids were assessed in simulated body fluid (SBF) and PBS, respectively. Cytocompatibility with mesenchymal stem cells was assessed using MTS and cell adhesion assays. Osteogenic differentiation was determined using the alkaline phosphatase activity (ALP), as well as the gene expression of Runx2 and Osterix markers. Hybrids loaded with 5, 10, and 15% of nBGs retained the mechanical flexibility of the PTHF–SiO_2_ matrix and improved its ability to promote the formation of bone-like apatite in SBF. The nBGs did not impair cell viability, increased the ALP activity, and upregulated the expression of Runx2 and Osterix. These results demonstrate that nBGs are an effective osteoinductive nanoadditive for the production of class II hybrid materials with enhanced properties for bone tissue regeneration.

## 1. Introduction

Bioactive glasses (BGs) are well known for their bone regenerative properties that promote the mineralization of bone-like apatite upon contact with physiological fluids and induce cells to differentiate into an osteogenic lineage [1,2]. However, glasses are brittle materials, a great drawback when used as solid blocks or scaffolds required to withstand cyclic loads [3]. These mechanical limitations have stimulated the development of a new class of hybrid biomaterials constituted by inorganic–organic phases interconnected at the molecular level by covalent bonds [4,5,6,7]. Hybrid materials prepared with high organic contents have demonstrated improved flexibility, comparable with thermoplastic polymers [4,8,9,10,11,12,13,14]. However, overcoming the brittleness issues of BGs [9,14,15,16,17,18,19,20,21,22] comes at the cost of a significantly lower bioactivity compared with pure BGs [4,23]. For instance, the immersion of gelatin–siloxane or chitosan–siloxane hybrids in simulated body fluid (SBF) did not produce bone-like apatite formation on the hybrid surface for three weeks [19,24]. Similarly, existing hybrid materials only modestly stimulate the expression of osteogenic-related gene markers such as phosphatase alkaline (ALP) activity [20]. To improve the bioactivity of hybrid materials, different chemical modifier agents have been incorporated into their structure. For instance, the addition of calcium nitrate has been shown to modify the structure of PMMA–silica [25] and gelatin–siloxane hybrids [19], resulting in increased ALP activity. Another approach consists of synthesizing hybrids with an inorganic phase formed by BG precursors. It has been shown that PCL- or PVA-based hybrid materials incorporated with BG increase the expression of osteogenic markers such as ALP, osteocalcin (OCN), and osteopontin (OPN) [20,21,22,26]. However, in most of the reported studies, hybrid materials were additionally incubated with cells in the presence of osteogenic supplements. On the other hand, Houaoui et al. [27] assessed microparticles of BG covalently linked to gelatin–siloxane hybrids through silane moieties, which showed improved material biodegradability, mineralization in SBF, and cell spreading and proliferation. However, the capacity of this microBG-modified hybrid material to stimulate the osteogenic cell differentiation process was not explored.

BG nanoparticles (nBGs) were found to exhibit superior bioactive properties to their microsized counterparts [28] due to their high surface area, resulting in a more rapid release of bioactive ionic dissolution products. Moreover, they have shown more favorable interfacial characteristics for the fabrication of nanocomposite materials. Several studies have demonstrated that nBGs accelerate the formation of bone-like apatite in vitro [4,29,30], induce a higher osteogenic cell differentiation [30,31,32,33], and enable a more rapid and complete bone tissue regeneration in vivo [29,30,32,34,35] when incorporated into polymeric scaffolds, titanium implant surfaces, or dental cements. However, the use of nBGs as nanometric agents to modify the bioactive properties of hybrid class II materials has not yet been reported. In principle, nBGs could be incorporated during the synthesis of hybrids in a controlled manner by the in situ synthesis of a nanocomposite hybrid material. This holds the prospect of combining the flexural mechanical properties of the hybrid matrix with the osteoinductivity of nBGs in a unique biomaterial. However, several challenges must be overcome to produce nBG-based hybrids with osteoinductive properties. The incorporation of ceramic nanofillers into hybrids may affect their mechanical flexibility [10,36,37], so the development of osteoinductive hybrids using nBGs as bioactive agents must be achieved without compromising their plasticity and toughness.

In this work, the synthesis of polytetrahydrofuran (PTHF)–silica (SiO_2_)-based hybrids modified with osteoinductive nBGs is explored. The composition of the nanocomposite hybrids was first optimized as a function of their mechanical flexibility. Then, the ability of the nBG-optimized hybrids to form bone-like apatite in SBF, their cytocompatibility, and their capacity to differentiate stem cells into an osteogenic lineage are assessed.

## 2. Materials and Methods

### 2.1. Synthesis of nBGs

nBGs were synthesized using the sol–gel method [31], with a molar composition of 58SiO_2_:40CaO:5P_2_O_5_. A calcium-based solution was prepared by dissolving Ca(NO_3_)_2_ × 4H_2_O in distilled water at room temperature. A second solution was prepared by diluting TEOS in ethanol. This was added to the calcium nitrate solution and the pH of the resulting solution was adjusted to 1–2 with nitric acid. This transparent solution was slowly dropped under vigorous stirring into a solution of NH_4_H_2_PO_4_ in 1500 mL of distilled water. During the dripping process, the pH was kept at around 10 with aqueous ammonia. The mixture was stirred for 48 h and aged for 48 h at room temperature. The precipitate was separated by centrifugation (12,000 rpm) and washed using three centrifugation–redispersion cycles with distilled water. This suspension was freeze-dried and then calcined at 700 °C for 3 h to obtain a fine white nBG powder.

### 2.2. Synthesis of Hybrid Materials

The hybrid materials were synthesized using tetrahydrofuran (THF) and silica (SiO_2_) as organic and inorganic phases, respectively, and (3-glycidyloxypropyl) trimethoxysilane (GPTMS) as a coupling agent. Nanocomposite hybrid materials were prepared by the incorporation of different amounts of nBGs (5.0, 10, and 15 wt.%) and tetraethyl orthosilicate (TEOS) (0.25, 0.5, and 1.0 mmol) (Table 1). The contents of the nBGs (wt.%) were estimated from the total mass of all the reaction constitutes. The concentrations of tetrahydrofuran (THF) and GPTMS were kept constant in all the studied synthesis compositions. In a typical synthesis, 450 µL (1.0 mmol) of TEOS was hydrolyzed into 100 µL of MilliQ Water, pH 2.0 (adjusted with 1M HCl), under stirring conditions and maintained at room temperature for 90 min. Subsequently, 16.2 mL of THF and 0.4 mL of GPTMS were mixed and combined with the dried nBGs.

The resulting suspension was stirred at room temperature for 30 min and then sonicated at 25 °C for 5 min. We added 80 µL of boron trifluoride etherate catalyst to the nBG–THF–GPTMS suspension, which was then stirred for 20 min before the dropwise addition of the hydrolyzed TEOS solution. The resulting solution was aged at room temperature for 60 min and then kept in an oven at 37 °C for 30 days to produce 3 cm^3^ cylindrical blocks of the hybrid material.

### 2.3. Material Characterization

#### 2.3.1. nBG Characterization

The morphology, composition, and particle size of the nBGs were analyzed using scanning electron microscopy (SEM) equipped with X-ray dispersive energy elemental microanalysis (EDX) in a JEOL-model microscope (JSM-IT300LV, Tokyo, Japan). In addition, particle size distribution was analyzed by dynamic light scattering (DLS) using Zeta Sizer Nano equipment (Malvern Instruments, Malvern, UK) at 25 °C with a scattering angle of 90° and a pH 7.4 nBG suspension of 1 mg/mL prepared in MilliQ water. nBGs were also analyzed for total attenuated reflectance by means of Fourier transform infrared spectroscopy (ATR-FTIR) using an Agilent Cary 630 ATR-FTIR spectrometer (Santa Clara, CA, USA), X-ray diffraction (XRD) using an X-ray diffractometer (STOE StadiP, Darmstadt, Germany), and atomic force microscopy (AFM) using a Nanosurf AG microscope (Gräubernstrasse Switzerland).

#### 2.3.2. Hybrid Material Characterization

The composition and microstructure of the hybrid materials were analyzed using SEM/EDX and the roughness and topography were characterized using AFM. The chemical structure of the hybrids was investigated using ATR-FTIR and solid-state nuclear magnetic resonance (NMR) spectroscopy (the experiments were carried out at 5.64 T using an Agilent DD2 spectrometer (Santa Clara, CA, USA), corresponding with a ^1^H Larmor frequency of 243 MHz). ^29^Si magic-angle spinning (MAS) NMR experiments were carried out at a Larmor frequency of 48.15 MHz in a 7.5 mm triple-resonance probe operating at a MAS frequency of υ MAS = 5 kHz. In total, 320 transients were collected using a 90° pulse of 8.5 μs duration and a recycle delay of 240 s. The ^29^Si chemical shifts were reported with respect to TMS (0 ppm). The degree of conversion (*Dc*) of the silica network could be calculated using Equation (1) [38], as follows:(1)$$Dc=\left(\left[\frac{{4Q}^{4}+{3Q}^{3}+{2Q}^{2}}{4}\right]+\left[\frac{{3T}^{3}+{2T}^{2}+T^{1}}{3}\right]\right)$$

The Q^n^ and T^n^ species correspond with Si(O-Si)_n_(OH)_4-n_ and C-Si(O-Si)_n_(OH)_3-n_, respectively [39]. Mechanical compression tests of 3 × 3 × 4 mm hybrid materials were performed using a DEBEN microtest machine (Suffolk, UK) with a 200 N load cell in compression mode. Elastic moduli were obtained as the mean value of the measurement of four specimens.

The mechanical properties of the hybrids were also analyzed by dynamic mechanical analysis (DMA) using DMA 8000 equipment (PerkinElmer, Shelton, CT, USA), where a sinusoidal deformation was applied to pieces with a geometry of 50 × 13 × 3 mm for 15 seconds in a temperature range of 20 to 50 °C. The thermal behavior of the materials was determined by differential scanning calorimetry (DSC) using DSC 8000 equipment (PerkinElmer). Heating scans at 10 °C/min using sealed T Zero pans in a nitrogen atmosphere in the temperature range of −40 to 160 °C were carried out. To determine the thermal stability of the materials, a thermogravimetric analysis (TGA) was conducted using a thermal analyzer (TA instruments, model Q50, New Castle, DE, USA). Therefore, ∼10 mg of material was heated over a temperature range of ∼20 to 800 °C at a heating rate of 10 °C min^−1^ in a nitrogen atmosphere with a flow rate of 40 mL min^−1^.

### 2.4. In Vitro Bioactivity Assays

The ability of the nanocomposite hybrids to induce apatite formation was assessed in acellular simulated body fluid (SBF) with inorganic ion concentrations comparable with those of human extracellular fluid. The SBF solution was prepared following the protocol of Kokubo et al. [40] using the standard ion composition (Na^+^ 142.0, K^+^ 5.0, Mg^2+^ 1.5, Ca^2+^ 2.5, Cl^−^ 147.8, HCO_3_^−^ 4.2, HPO_4_^2−^ 1.0, and SO_4_^2−^ 0.5 mM). The fluid was buffered at a physiological pH of 7.4 at 37 °C with tri-(hydroxymethyl) aminomethane and hydrochloric acid. Blocks of 2 × 1 × 1 cm hybrid materials were individually soaked in 50 mL SBF inside polyethylene containers at 36.5 °C using a thermostatic water-bath shaker. After incubation for 7 and 14 days, the hybrids were removed from the SBF, rinsed with distilled water, and dried at 50 °C.

### 2.5. Hybrid Degradation Test

The degradability of the materials was assessed by immersion in a phosphate buffer (PBS, pH 7.4) for 28 days. Specimens of 5 × 5 × 5 mm were frozen at −20 °C and dried in a lyophilizer (ilshin BioBase, Dongducheon, Republic of Korea) at −60 °C for 24 h. The weight loss after immersion in PBS at pH 7.4 and 37 °C for 28 days was recorded using an analytical balance. The material degradation was calculated from the normalized difference of the initial weight *W*_0_ and the weight *W*_t_ after the desired time *t* according to Equation (2) [41], as follows:(2)$$Degradation\%=\left(\frac{W_{0}-W_{t}}{W_{0}}\right)\times 100\%$$
where *W*_0_ is the initial weight of the scaffold and *W_t_* is the weight of the scaffold at the respective time point.

### 2.6. Cell Culture

#### 2.6.1. Cytocompatibility Assays

Stem cells isolated from human dental pulp (hDPSCs) were used to evaluate cell proliferation and differentiation. hDPSCs were cultured in Dulbecco’s Modified Eagle Medium (DMEM; Invitrogen Life Technologies, Waltham, MA, USA) containing 10% fetal bovine serum (FBS GIBCO, Carlsbad, CA, USA), 100 U/mL penicillin, and 100 mg/mL streptomycin; 5 × 10^4^ cells were directly seeded onto 5 × 5 × 5 mm cubic samples of hybrid material and placed in a single well of a 48-well cell-culture plate. Cell viability was determined after 3 and 7 days of incubation by using the CellTiter 96^®^ AQueous One Solution Cell Proliferation Assay (Promega, Madison, WI, USA), which measures the reduction of [3–(4,5-dimethylthiazol-2-yl)-5–(3-carboxymethoxyphenyl)-2–(4-sulfophenyl)-2H-tetrazolium] (MTS) to formazan by mitochondria in viable cells. After 2 h of incubation with the MTS reagent incubated at 37 °C in a humidified air atmosphere containing 5% CO_2_, the medium was collected from the samples and absorbance was measured at a wavelength of 490 nm using an ELISA microplate reader (Tecan Infinite F-50). Cell adhesion on the hybrid surfaces was also assessed. The cells that had adhered to the surface of each nanocomposite hybrid after 24 h of incubation were examined using SEM. For this purpose, DPSC cells were fixed in 2.5% glutaraldehyde, progressively dehydrated in ethanol, dried in super-critical CO_2_, and finally coated with gold.

#### 2.6.2. Cell Differentiation Assays

The capacity of the hybrid materials to induce cell differentiation into an osteogenic lineage was assessed using different biomarkers in the absence of osteogenic supplements. hDPSCs were cultured in Dulbecco’s Modified Eagle Medium (DMEM; Invitrogen Life Technologies) containing 10% fetal bovine serum (FBS, GIBCO), 100 U/mL penicillin, and 100 mg/mL streptomycin; 5 × 10^4^ cells were directly seeded onto 5 × 5 × 5 mm cubic samples of hybrid material placed in a single well of a 48-well cell-culture plate in triplicate and cultured with the hybrids.

The ability of the hybrids to produce an osteogenic differentiation of DPSCs was assessed in the absence of osteogenic supplements after 7 and 14 days of incubation by measuring the activity of the alkaline phosphatase (ALP) and the gene expression of Runx2 and Osterix. The activity of the alkaline phosphatase (ALP) enzyme was determined by the colorimetric dephosphorylation assay of the p-nitrophenyl phosphate reagent at 405 nm. To analyze the Runx2 and Osterix gene expression, the materials were incubated with hDPSCs for 3 days for Runx2 and 14 days for Osterix. Total RNA was isolated from the control cells and those treated with the conditioned medium using Trizol (GIBCO, Carlsbad, CA, USA). cDNA was generated using the ReadyScript cDNA Synthesis Mix (Sigma). Real-time quantitative PCR reactions were obtained using the following human-specific primers (Macrogen): RUNX-2 forward 5′-CAAGTA- GCAAGGTTCAACGA-3′ and reverse 5′-CGGTCAGAGAACAAACTAGG-3′; OSX forward 5′-GCCAGAAGCTGTGAAACCTC-3′ and reverse 5′-TGATGGGGTCATGGTGTCTA-3′; and 18S forward 5′-GGACACGGACAGGATTGACA-3′ and reverse 5′-GGACATCTAAGGGCATCACAG-3′. The expression of 18S was analyzed as a loading control. Quantitative PCR was performed using a StepOnePlus^TM^ Real-Time PCR system (Applied Biosystems). Each reaction was conducted using MicroAmp^®^Fast Reaction Tubes (Applied Biosystems) with 100 ng of cDNA at a final volume of 10 µL. The PCR mixture contained Power SYBR^®^Green PCR MasterMix (Applied Biosystems) and 500 nM of each primer (forward and reverse). Fluorescence was analyzed using StepOnePlus software, version 2.3 (StepOnePlus Real-Time PCR, Life Technology, Carlsbad, CA, USA). The quantification of the gene expression was determined through the fold-change relative to the control condition.

## 3. Results and Discussion

### 3.1. nBG Characterization

Figure 1A shows a representative SEM image of the synthesized nBG nanopowder. The particle size histogram (Figure 1B) elaborated from the SEM measurements shows a size range from 50 to 140 nm, with a mean size of ca. 80 nm. The mean particle size analyzed using DLS (Figure 1C) was 108 nm, a value consistent with that estimated from SEM observations considering the particle aggregation effects associated with DLS measurements.

### 3.2. Mechanical Properties of Synthesized Hybrid Materials

The synthesis products obtained from the compositions under study were analyzed regarding their mechanical flexibility (Figure 2; Appendix A).

Figure 2A reveals that certain hybrid nanocomposites did not fracture under the applied compression conditions, most notably the hybrid composition 0.25TEOS. Three hybrids formulated with nBGs (5nBG/0.5TEOS, 10nBG/1TEOS, and 15nBG/0.5TEOS) exhibited mechanical flexibility. The effect of the combination of different contents of nBGs and concentrations of TEOS (SiO_2_ content) on the flexibility of the material was complex and non-linear. It appeared that the hybrids with intermediate silica concentrations rendered the composites containing larger nBG contents more flexible. The flexibility of hybrids depends on several factors, such as their molecular homogeneity and concentration as well as the molecular weight of the silicate/organic oligomers, crosslinks between the inorganic and organic phases, interactions between particle nanofillers, and hybrid network, among others [7]. Although nBG incorporation tended to increase the compressive strength and elastic modulus of some hybrids, the flexible materials did not differ in those properties (Figure 2B,C). The DMA measurements obtained from the selected hybrids (Figure 2D) showed that their storage moduli (G′) were lower than their loss moduli (G″) (Tan δ < 1), which confirmed a major contribution of the elastic component to the mechanical behavior in accordance with their exhibited flexibility. In addition, hybrids loaded with nBGs presented G′ and G″ values higher than those of the 0.25TEOS hybrid, suggesting a reinforcement effect of the nBGs in the hybrid matrix. It is known that nanoparticles with high surface-area-to-volume ratios restrict the mobility of polymer chains [42,43], producing an increase in the storage modulus [44]. Moreover, a reduction in the damping factor (Tan δ) with the incorporation of nBGs was observed, likely due to the increase in G″ values as consequence of increased frictional forces and heat dissipation [45].

As 0.25TEOS, 5nBG/0.5TEOS, 10nBG/1TEOS, and 15nBG/0.5TEOS were the only hybrids that exhibited mechanical flexibility, these materials were selected for structural characterization and an assessment of their bioactive properties.

### 3.3. Structural Characterization of Selected Hybrids

Figure 3 shows the FTIR spectra of the selected hybrids and nBGs. The spectra of PTHF and silica (SiO_2_) are also included as a reference. The assignment of the main FTIR vibrations exhibited by the material spectra are summarized in Table 2.

The spectra of the hybrids presented characteristic bands associated with vibrations of the Si–O–Si bonds in the siliceous network (~540, ~880, and 1100–1000 cm^−1^) as well as some of the vibrations produced by PTHF polymer chains (2800, 2900, 1365, and 740 cm^−1^). The FTIR bands of the nBGs were not easily observed in the nanocomposite hybrid spectra because the siliceous vibrations of the nanoparticles were not distinguishable from the vitreous bands of the hybrid matrix and overlapped with some of the PTHF vibrations. In addition, the hybrids exhibited a band of relatively high intensity at 1110–1080 cm^−1^, which has been attributed to the asymmetric vibrational motions of Si-O-C moieties in sol–gel-derived siloxane hybrid materials [46,56,57]. This covalent bond between the polymer segments and silica network confirmed the formation of a class II hybrid material [15,47,57]. To further characterize the connectivity of the silica network and its degree of condensation, we carried out ^29^Si MAS NMR experiments (Figure 4).

The spectra showed a number of signals over a chemical shift range of about −50 to −120 ppm, attributable to T^n^ and Q^n^ units corresponding with CSi(OSi)_n_(OR)_3-n_ and Si(OSi)_n_(OR)_4-n_ species (R:H, CH_3_CH_2_ or O^−^), respectively, present in the synthesized materials. A deconvolution of the ^29^Si MAS NMR spectra into Gaussian components using least squares fitting allowed the extraction of the NMR parameters summarized in Table 3. The dominant presence of Q^4^ units confirmed that the silica component of the hybrids condensated into a three-dimensional network in both the hybrid and nanocomposite hybrid materials. Moreover, the 15nBG/0.5TEOS hybrid contained a Q^4b^ species produced by silicon nuclei in a significantly different local environment [58,59]. Although Q^3^ and Q^2^ units, related to partially condensed silicate units, were also detected, the degree of condensation (% Dc) increased in the hybrids loaded with 10 and 15% wt. of nBGs as a consequence of the highly condensed silica network of the sol–gel-derived nanoparticles and higher TEOS concentration of the synthesis mixture [60,61]. The NMR spectra also exhibited T^n^ units that confirmed the presence of Si-O-C bonds. The decrease in the abundance of less condensed T^1^ species with nBGs and TEOS content also correlated well with the increase in the % Dc. Thus, the nanocomposite hybrids formulated with nBGs presented a more condensed silica structure compared with that of a hybrid without nBGs.

The incorporation of nBGs into the hybrid matrices are visually noted in Figure 5A by the decrease in the translucency of the materials as the content of nanoparticles increased. A surface examination of the materials using SEM revealed the presence of some nBG particles embedded in the matrix of the hybrid nanocomposites (Figure 5B). The intensity of the EDX maps of Si suggested a relatively homogeneous distribution of nBGs into the hybrid. Likewise, the concentrations of Si, Ca, P, and O was higher as the percentage of nBGs incorporated into the hybrids increased (Figure 5C). AFM further provided evidence of the incorporation of the nanoparticles into the hybrid and 3D topographic images showed that the nBG-loaded hybrids had a coarser surface, with a significant increase in the root mean square roughness (Rq) from approximately 1 to 300 nm (Figure 5D). It is known that the roughness of a biomaterial is an important feature that promotes cell adhesion, activating signaling pathways that regulate the stages of cell proliferation, migration, and differentiation in the process of bone tissue regeneration [62,63,64]. Figure 5E also shows the XRD patterns of the nBGs and hybrid materials. The XRD analysis confirmed the amorphous nature of the sol–gel synthesized nBGs, whereas the diffractogram of the 0.25TEOS hybrid showed reflections at 20 and 25° associated with the semicrystalline structure of PTHF [65,66,67]. Interestingly, the peaks from the semicrystalline moieties were lower in the 5nBG/0.5TEOS material and were completely absent in 10nBG/1TEOS and 15nBG/0.5TEOS. This effect has been also observed in polymer nanocomposites, attributable to a disruption in the spherical semicrystalline regions (spherulites) of the polymer by nBG particles [68,69,70]. Thus, the data suggested that the incorporation of nBGs into the PTHF–SiO_2_ system produced more amorphous hybrid materials.

It is known that nanofillers can increase the thermal stability of a polymer matrix through barrier effects, specific interactions, and restricted chain mobility [71,72]. In the present case, it was found that the thermal stability of the hybrid matrix was affected by the presence of the nanoparticles. Figure 6A shows that the maximum degradation rate of the nBG-loaded materials shifted with higher temperatures when compared with the 0.25TEOS hybrid.

The degradability of the hybrid materials in PBS was also assessed (Figure 6B). Hybrids loaded with nBGs exhibited a larger weight loss compared with the 0.25TEOS hybrid. Moreover, the degradability of the nanocomposite was larger for the hybrids prepared with the 0.5TEOS matrix (5nBG/0.5TEOS and 15nBG/0.5TEOS) than those formulated with the 1TEOS matrix. Our nanocomposites based on the 0.5TEOS/PTHF hybrid matrix combined with nBGs presented weight loss of around 8–10 wt.% after 28 days of immersion in PBS, which was comparable with the degradability (6 to 8 wt.%) reported for hybrids formulated with more degradable polymer matrices such as TEOS/PCL [73] and TEOS/PCL/PTHF [74]. Considering that PTHF is a stable polymeric phase [75,76], the weight loss exhibited by our nBG/TEOS/PTHF hybrids was preferably attributed to the degradation of the silica/nBG inorganic phase, particularly the partial dissolution of the reactive nBG particles [77,78]. Thus, the incorporation of nanosized nBGs into the hybrids could improve the degradability of these biomaterials, as already observed in polymer-based nanocomposite scaffolds [29,79].

### 3.4. In Vitro Bioactivity in SBF

The ability of the hybrids to form bone-like apatite on their surface was assessed through immersing the materials in SBF for 7 and 14 consecutive days. XRD diffractograms of the hybrid surfaces (Figure 7) did not present reflections associated with the structure of apatite (particularly the more intensive peak at 31.7° 2θ). Although XRD is undeniably a central characterization tool for the identification of apatite, in the case of the SBF immersion test, apatite crystalline deposits of very limited depth can remain, making their detection difficult [80]. In addition, XRD detection is affected when a thin and poorly crystalline apatite layer is formed or with preferential crystal orientations. FTIR is a well-validated surface analysis method used to detect the crystallization of the apatite layer on the surface of bioactive glass-based biomaterials [81,82]. ATR-FTIR spectra of the hybrids before and after their immersion in SBF are shown in Figure 8. Infrared spectroscopy is sensitive to the formation of apatite, showing a double signal around 550–600 cm^−1^ attributed to the P–O bending vibration and a band at 1000–1100 cm^−1^ associated with the P–O symmetric stretching vibration in crystalline apatite.

The presence of apatite vibrations could be observed in the spectra of all hybrids after immersion in SBF, especially after 14 days, without any significant differences among them. The surfaces were further analyzed using SEM and EDX elemental analyses (Figure 9).

The SEM images confirmed the formation of mineral deposits on the surface of the hybrids after immersion in SBF. Although apatite crystals ideally exhibit a tiny flake-like morphology, apatite produced on bioactive glass surfaces is commonly observed as circular and with non-regular particles, which may be a consequence of Ca-deficient apatite precipitation [31,83,84]. Our SEM images showed a higher degree of mineralization in terms of the density and cluster size of apatite on the hybrids loaded with nBGs, especially after 14 days of immersion in SBF. In addition, EDX analysis of the mineralized surfaces revealed an increased concentration of Ca and P with an increase in nBG content, which was consistent with the surface transformation of bioactive glass into the apatite phase [85]. It has already been demonstrated that nBGs improve the ability of a material to induce apatite formation when incorporated into titanium implants [86], polymer scaffolds [32], dental cements [87], and hydrogels [88]. nBGs induce more rapid apatite crystallization than traditional microsized BGs due to their nanometric dimensions, larger surface area, and, consequently, higher rate of dissolution [89]. Several strategies have been tested to produce hybrids with an apatite-forming ability comparable with that of inorganic bioactive glasses. Rhee et al. [90] enhanced the formation of apatite on SiO_2_–PCL hybrids by incorporating calcium nitrate in the sol–gel synthesis mixture. Koh et al. [91,92] synthesized hybrids of poly (tetramethylene oxide) combined with triethyl phosphate, calcium chloride, and TEOS that formed apatite, depending on the calcium and phosphate ion concentrations dissolved from the hybrids. Mondal et al. [92] prepared a hybrid through the condensation of PCL with a borophosphosilicate network, which exhibited the ability to deposit apatite when incubated in SBF. In the current study, the incorporation of nBGs into the hybrid structure appeared to be an attractive strategy to improve the apatite-forming ability of less reactive hybrid matrices as the content of the bioactive nanoparticles could easily be tuned to produce the desired bioactivity.

### 3.5. Cytocompatibility and Osteogenic Differentiation

The cell viability of hDPSCs cultured with the hybrids was assessed after 3 and 7 days of incubation (Figure 10A).

The results of the MTS assay demonstrated that the viability of the hDPSCs incubated with the hybrids did not significantly differ from that of the control cells cultured without materials. Moreover, SEM (Figure 10B–E) showed that cells intimately adhered to the hybrid surfaces and developed lamellipodia and filopodia extensions on the substrate. The excellent cytocompatibility exhibited by the hybrid materials was not impaired by their content of nBGs as the latter have been shown to be cytocompatible and even increase cell proliferation due to calcium release effects and high surface energy [31,93,94]. The osteoinductive capability of the hybrids was also assessed in the absence of osteogenic supplements. Figure 11 shows that the activity of ALP in the DPSCs incubated with the hybrids significantly increased when compared with the cells cultured without materials (control).

In particular, the highest activity of ALP was observed when cells were incubated with the 15nBG/0.5TEOS hybrid. An ALP enzyme is produced when osteoblasts lay down the bone extracellular matrix. Consequently, it was a clear marker of the osteogenic cell differentiation process promoted by the hybrids [95]. Moreover, hDPSCs cultured with hybrids modified with 10 and 15% nBGs had significantly higher expression levels of the osteogenic transcription factors Runx2 and Osterix than the control cells and 0.25TEOS hybrid (Figure 12).

The expression of these markers promoted by the nanocomposite hybrids was somewhat lower than that produced by the nBG nanopowder and the osteogenic medium.

The development of osteoinductive properties in hybrid materials has been relatively less studied. Most studies focus on the assessment of cytotoxicity and cell adhesion of the hybrids [9,14,27] or the exploration of cartilage regeneration applications [15]. To develop hybrids with osteoinductive properties, the most targeted approach has been the incorporation of calcium ions into the hybrid structure. Poly (methyl methacrylate)/silica [19], chitosan/silicate [96], and polyethyleneimine (PEI)/bioactive glass [97] hybrids modified with calcium salts have been demonstrated to increase the ALP activity and other osteogenic markers in differentiated osteoblasts. Other approaches, such as chitosan–silicate hybrids enriched with silicon, have also enhanced the ALP activity [98] and hybrids of PCL condensed with borophosphosilicate glass upregulated the expression of ALP, osteopontin (OPN), and osteocalcin (OCN), depending on the boron concentration [21]. In our nanocomposite hybrid, the presence of nBGs was the factor that promoted the osteoinductive properties. It has been well demonstrated that the ionic dissolution products of BGs activate the osteoblast genotype expression [99,100,101], an effect that is accelerated by the nanometric size and high surface-area-to-volume ratio of nBGs [102,103,104]. Figure 13 shows the chemical reactions involved in the synthesis of the nanocomposite hybrid as well as a schematic representation of a proposed structure indicating the possible intermolecular interactions in the material. The reaction began with the silanization of the THF with GPTMS and cationic ring-opening polymerization in the presence of nBGs. Then, the sol–gel condensation of GPTMS–PTHF with the hydrolyzed TEOS and the polymer chain growth took place. The hybrid SiO_2_–PTHF matrix exhibited the typical covalent bonds of class II hybrids, where organic and inorganic chains were bound through the GPTMS silane coupling agent. Weaker London and dipole–dipole interactions also occurred between the silica and polymer components. On the other hand, nBGs were distributed within the hybrid matrix, which may have interacted with both the silica components of the hybrid and neighboring nanoparticles through hydrogen bonding and weaker forces. Thus, the partial erosion of the hybrid matrix could produce the exposition of the nBGs to the medium and the consequent release of soluble ions that promote surface apatite mineralization and activate the osteogenic cell differentiation process. The in vitro osteoinductive properties exhibited by the nBG-modified hybrids are expected to reduce the time required for bone tissue reconstruction in vivo and may produce a tissue of higher quality, as has already been demonstrated when polymer scaffolds are loaded with nBGs [29,32,88]. In the future, in vivo experiments are necessary to verify the bone regenerative properties of this new kind of hybrid material formulated with osteoinductive nBG particles. In addition, different polymer–silica combinations can be explored to produce nanocomposite hybrids with tuned osteogenic properties.

## 4. Conclusions

The results of this study demonstrate that it is possible to incorporate nBGs into a PTHF–SiO_2_-based class II hybrid matrix to improve its in vitro osteogenic properties while retaining its mechanical flexibility. The nanoparticles enhanced the ability of the hybrid to promote apatite formation in SBF, did not impair cell viability, and upregulated the expression of osteogenic differentiation markers.

## Figures and Tables

**Figure 1 biomolecules-14-00482-f001:**
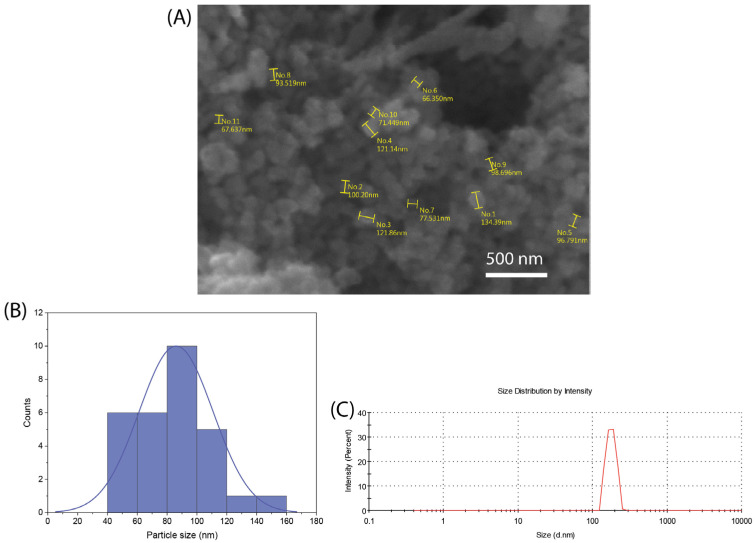
SEM image (**A**), particle size distribution histogram obtained from SEM measurements (**B**), and DLS particle size analysis curve (**C**) of synthesized nBG powder.

**Figure 2 biomolecules-14-00482-f002:**
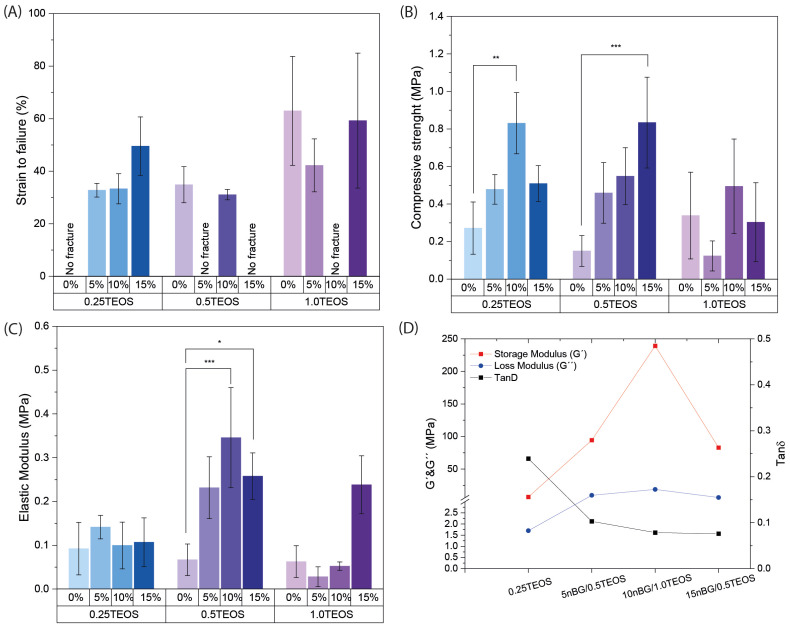
Mechanical properties of hybrids: strain to failure (**A**), compressive strength (**B**), elastic modulus (**C**), and DMA measurements of the storage modulus (G′), the loss modulus (G″), and the damping factor Tanδ (**D**) (* *p* ≥ 0.05; ** *p* ≥ 0.01; *** *p* ≥ 0.001).

**Figure 3 biomolecules-14-00482-f003:**
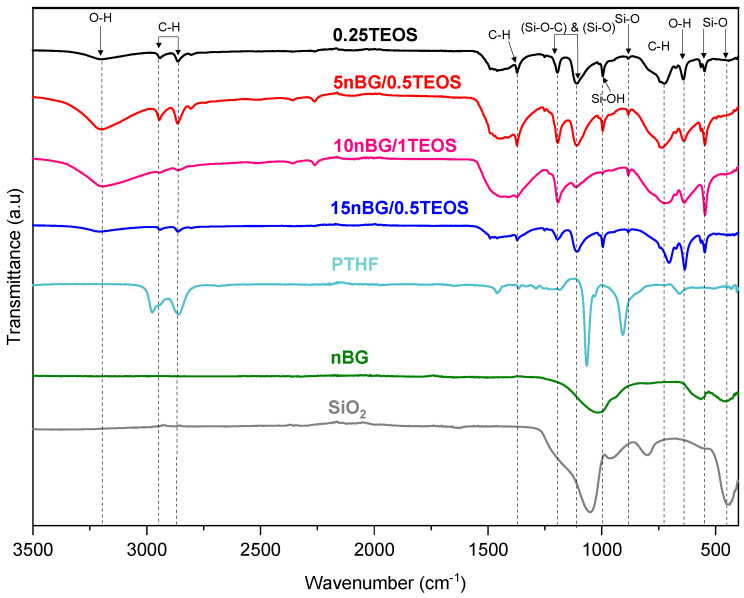
FTIR – ATR spectra of selected hybrid materials and nBGs. Spectra of PTHF and silica (SiO_2_) are included as a reference.

**Figure 4 biomolecules-14-00482-f004:**
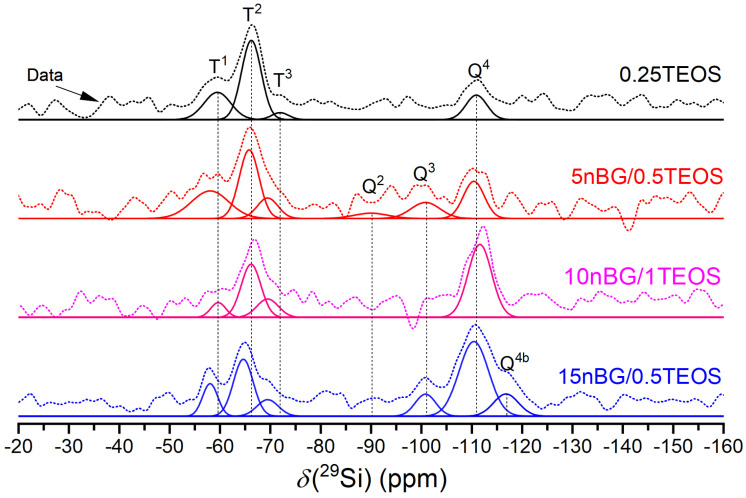
^29^Si MAS NMR spectra of the materials under study. The thick solid line represents the experimental spectrum. The other solid lines are the Gaussian components of the silicate units and the deconvolutions.

**Figure 5 biomolecules-14-00482-f005:**
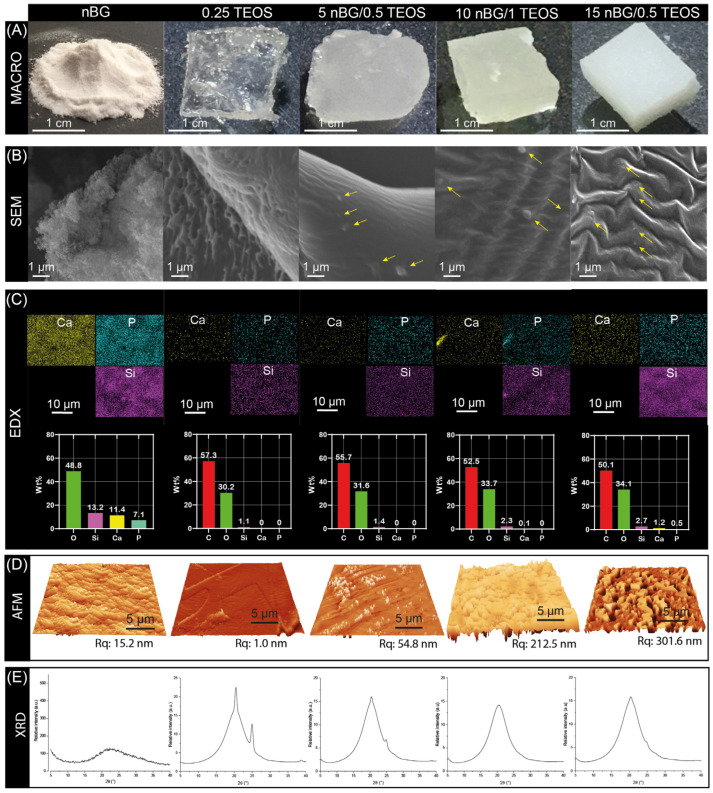
Macro photographs (**A**), SEM images (arrows showing the embedded nBG particles) (**B**), EDX elemental analysis (**C**), AFM 3D views with Rq values (**D**), and XRD patterns (**E**) of nBGs and hybrid materials.

**Figure 6 biomolecules-14-00482-f006:**
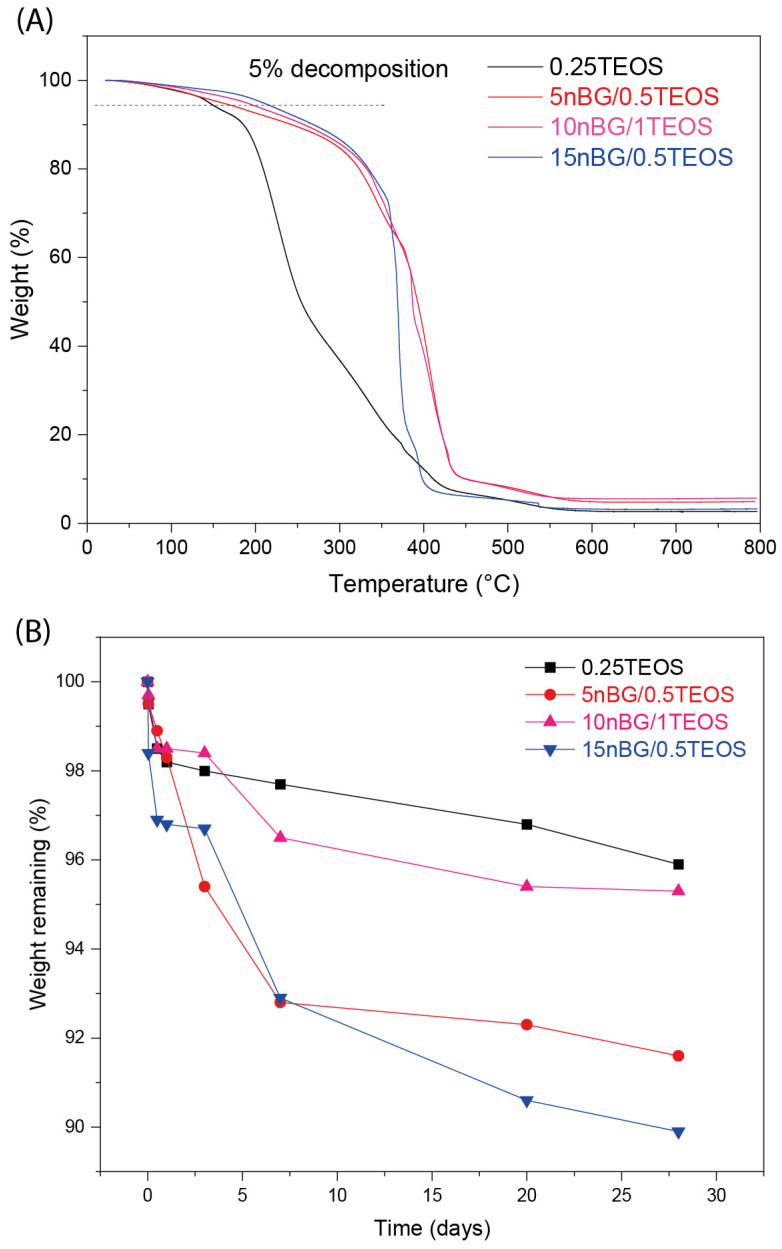
TGA analysis (**A**) and weight loss of the hybrid materials after 28 days of immersion in PBS pH 7.4 at 37 °C (**B**).

**Figure 7 biomolecules-14-00482-f007:**
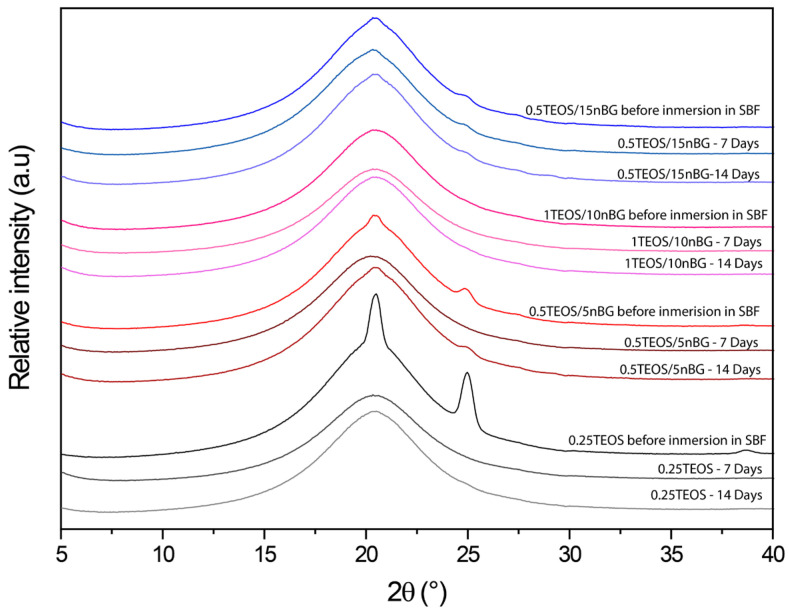
XRD patterns of hybrids before and after being conditioned in SBF at 37 °C for 7 and 14 days.

**Figure 8 biomolecules-14-00482-f008:**
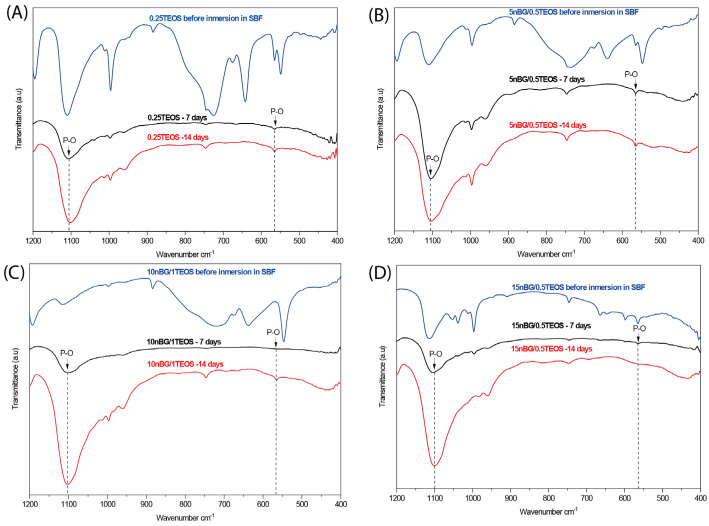
ATR – FTIR spectra of hybrids before and after being conditioned in SBF at 37 °C for 7 and 14 days: (**A**) 0.25TEOS, (**B**) 5nBG/0.5TEOS, (**C**) 10nBG/1.0TEOS, and (**D**) 15nBG/0.5TEOS.

**Figure 9 biomolecules-14-00482-f009:**
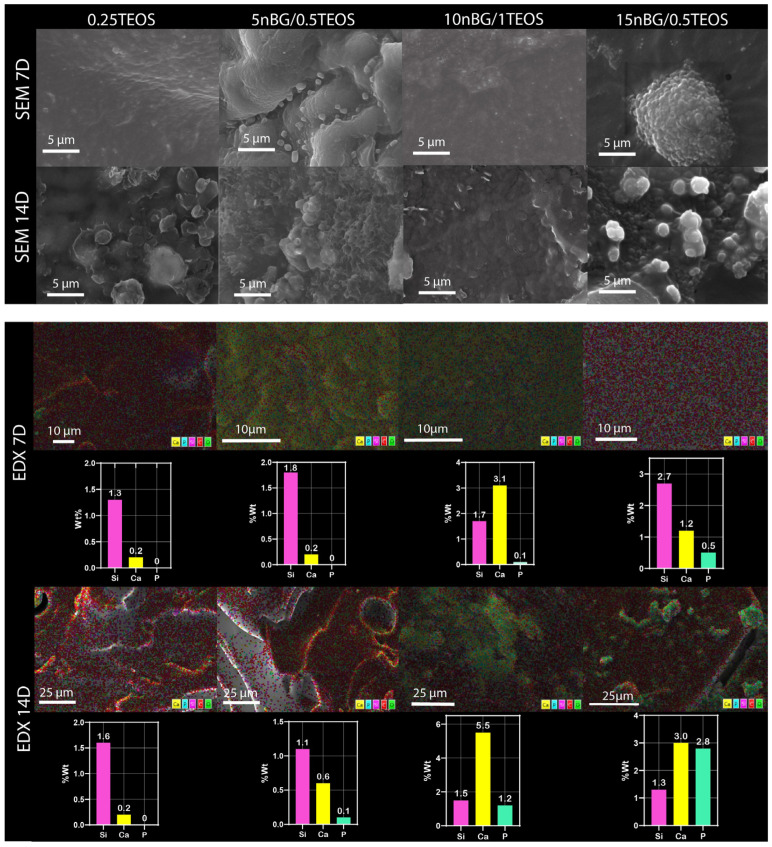
Surface SEM images, elemental mapping EDX, and content of silicon, calcium, and phosphorus of nanocomposite hybrids after conditioning in SBF for 7 and 14 days.

**Figure 10 biomolecules-14-00482-f010:**
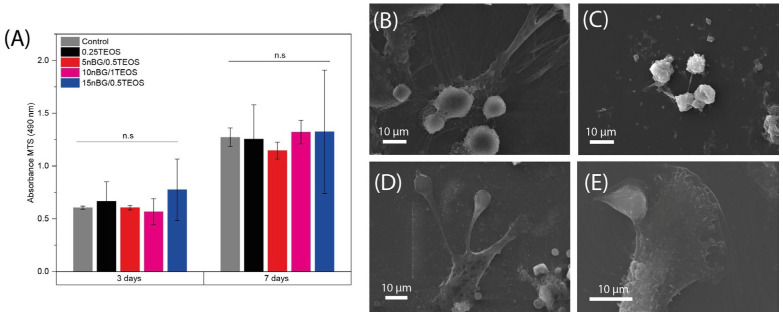
Viability of DPSCs cultured in the presence of nanocomposite hybrids measured using MTS assay after 3 and 7 days of incubation (**A**), and SEM images of cells adhered on 0.25TEOS (**B**), 5nBG/0.5TEOS (**C**), 10nBG/1TEOS (**D**), and 15nBG/0.5TEOS (**E**) hybrid surfaces after 48 hours of incubation (n.s: non-significance; *p* > 0.05).

**Figure 11 biomolecules-14-00482-f011:**
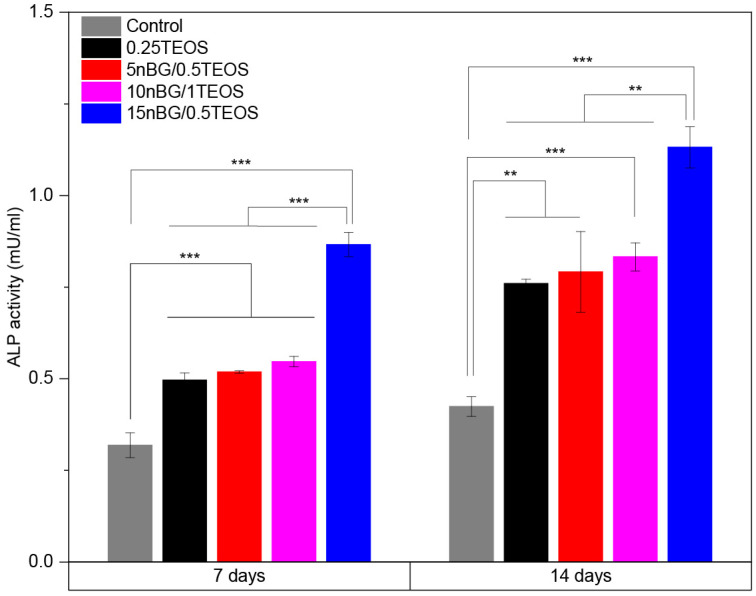
ALP activity of DPSCs incubated with nanocomposite hybrids for 7 and 14 days in the absence of osteogenic supplements (significance levels: * *p* < 0.05, ** *p* < 0.01, and *** *p* < 0.001).

**Figure 12 biomolecules-14-00482-f012:**
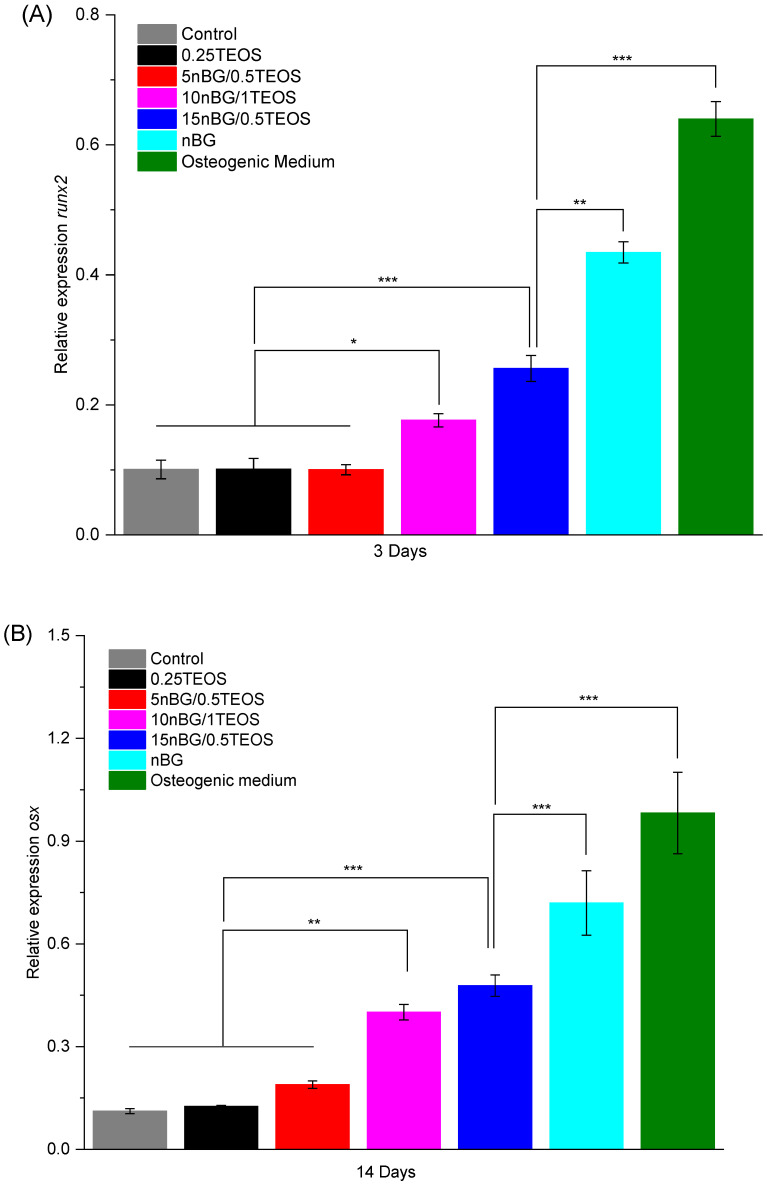
Relative expression of Runx2 (**A**) and Osterix (**B**) osteogenic markers in DPSCs incubated with nanocomposite hybrids for 3 and 14 days in the absence of osteogenic supplements. Positive controls are DPSCs cultivated with osteogenic medium and bioactive glass nanoparticles (significance levels: * *p* < 0.05, ** *p* < 0.01, and *** *p* < 0.001).

**Figure 13 biomolecules-14-00482-f013:**
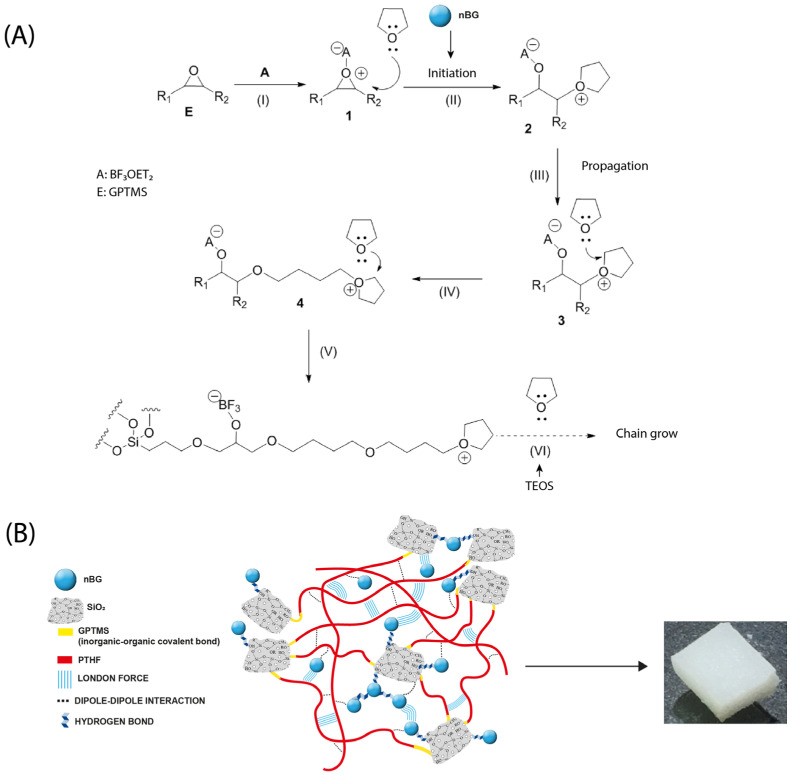
Chemical reactions involved in the synthesis of PTHF–SiO_2_ nanocomposite hybrid loaded with bioactive glass nanoparticles (nBGs) (**A**) and proposed structure of the material indicating the possible intermolecular interactions (**B**).

**Table 1 biomolecules-14-00482-t001:** Studied compositions for the synthesis of hybrids and nanocomposite hybrids.

Sample	nBGs (wt.%)	TEOS (mmol)
0.25TEOS	-	0.25
5nBG/0.25TEOS	5	0.25
10nBG/0.25TEOS	10	0.25
15nBG/0.25TEOS	15	0.25
0.5TEOS	-	0.50
5nBG/0.5TEOS	5	0.50
10nBG/0.5TEOS	10	0.50
15nBG/0.5TEOS	15	0.50
1TEOS	-	1.00
5nBG/1TEOS	5	1.00
10nBG/1TEOS	10	1.00
15nBG/1TEOS	15	1.00

* The concentrations of THF and GPTMS were kept constant. The weight percentage of nBGs was ascertained from a hybrid without nBGs.

**Table 2 biomolecules-14-00482-t002:** Assignment of the main FTIR signals shown by the spectra of hybrid materials, nBGs, PTHF, and silica.

Peak Position (cm^−1^)	Assignment	References
3200	Asymmetrical stretch –OH	[15,46]
2800–2900	Stretch CH_2_	[15,46]
1375–1365	Stretch C-H	[15,46]
1110–1080	Stretch Si-O-C	[15,46,47,48,49,50]
1100–1000	Asymmetrical stretch Si–O	[15,46,47,51]
990–950	Stretch Si–OH	[15,47,50]
880–800	Symmetrical stretch Si–O	[15,52]
720–700	Cis Movement C-H	[46,51]
~634	Release mode –OH	[53,54,55]
~540	Si–O–Si asymmetrical bending	[15,47]
~460	Si–O–Si bending	[15,47]

**Table 3 biomolecules-14-00482-t003:** ^29^Si isotropic chemical shifts (*δ*_iso_) and percentage abundance (*f*) of silicon T and Q species and *Dc* percentage of hybrids.

Samples	Q^2^		Q^3.^		Q^4^		Q^4b^		T^1^		T^2^		T^3^		%D_c_
	*δ*_iso_ (ppm)	*f* %	*δ*_iso_ (ppm)	*f* %	*δ*_iso_ (ppm)	*f* %	*δ*_iso_ (ppm)	*f* %	*δ*_iso_ (ppm)	*f* %	*δ*_iso_ (ppm)	*f* %	*δ*_iso_ (ppm)	*f* %	
0.25TEOS	-	-	-	-	−110.9	18	-	-	−59.6	25	−66.2	53	−71.9	4	65.67
5nBG/0.5TEOS	−90.0	5	−100.8	11	−110.4	18	-	-	−58.0	24	−65.8	32	−69.5	10	68.08
10nBG/1TEOS	-	-	-	-	−111.6	51	-	-	−59.6	6	−66.2	31	−69.5	12	85.67
15nBG/0.5TEOS	-	-	−100.8	9	−110.4	42	−116.8	11	−58.0	9	−64.6	22	−69.5	7	84.42

*δ*_iso_ represents the ^29^Si chemical shift. Errors associated with measurements are ± 1.0 ppm. *f* % represents the percentage abundance (*f*) of silicon T and Q species of hybrids. %Dc represents the degree of condensation in the silica network. Errors associated with measurements of *f* and Dc are ± 5%.

## Data Availability

Data are contained within the article.

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
