# Peer review of "Novel Organic–Inorganic Nanocomposite Hybrids Based on Bioactive Glass Nanoparticles and Their Enhanced Osteoinductive Properties"

_biomolecules, 2024, doi:10.3390/biom14040482_

Round 1

Reviewer 1 Report

Comments and Suggestions for Authors

Generally, the manuscript reads well and is nicely organized. The manuscript can be accepted for publication, following minor revisions as suggested following:

1. The authors define nBG twice, once in the Introduction and then in Materials & Methods. The abbreviation can be used the second time without again defining it.

2. Section 2.3 Material characterization includes several tests performed. They should be sub-sectioned as 2.3.1 and so on, as done for section 2.5.

3. Result sections 3.1 and 3.2 describe the results of methods described in section 2.3. The titles for the sections can be matched for swift understanding and a good flow of the reading.

4. Fig 4 details the results of SEM, EDX, XRD, and AFM; the text below the figure discusses the results shown in the figure, but they don't refer to the image till they talk about XRD. It will help to number the images within Fig 4 as 4a, 4b, and so on to make the reader understand clearly.

5. It is unclear what Fig 8D and 8E are. If they are not talked about in the text, they can be removed, OR they should be referred to in the text and detailed in the caption.

6. The captions of the figures can be improved by providing more details.

Comments on the Quality of English Language

The authors have used 'neat' for the control bioceramic materials that are not hybrid. I encourage the authors to use the term 'bioceramics' OR choose another word instead of 'neat.' It is not a scientific word.

Other than that, minor English editing will polish the manuscript.

Author Response

We thank reviewers for their valuable comments, observations, and corrections to improve the quality of our manuscript. In the elaboration of the revised version of the manuscript, all the observations made by the reviewers were carefully considered. The changes made in the manuscript text appear highlighted in colored letters to better clarity. In addition, replies to each reviewer’s comments are indicated below.

Reviewer 1.

Generally, the manuscript reads well and is nicely organized. The manuscript can be accepted for publication, following minor revisions as suggested following:

  1. The authors define nBG twice, once in the Introduction and then in Materials & Methods. The abbreviation can be used the second time without again defining it.

- nBG is now defined only in the introduction.

  1. Section 2.3 Material characterization includes several tests performed. They should be sub-sectioned as 2.3.1 and so on, as done for section 2.5.

- Section 2.3 was now sub-sectioned.

  1. Result sections 3.1 and 3.2 describe the results of methods described in section 2.3. The titles for the sections can be matched for swift understanding and a good flow of the reading.

- The titles of the result sections 3.1 and 3.2 were rewritten to match with those of the method section.     

  1. Fig 4 details the results of SEM, EDX, XRD, and AFM; the text below the figure discusses the results shown in the figure, but they don't refer to the image till they talk about XRD. It will help to number the images within Fig 4 as 4a, 4b, and so on to make the reader understand clearly.

- Images of current Fig. 5 (former Fig. 4) were identified as Fig. A, B, C, D, and E depending on the type of analysis.

  1. It is unclear what Fig 8D and 8E are. If they are not talked about in the text, they can be removed, OR they should be referred to in the text and detailed in the caption.

- Fig. 8B, 8C, 8D, and 8E are now identified in the caption.

  1. The captions of the figures can be improved by providing more details.

- The captions of the figures were improved throughout the manuscript by adding more detailed information.

  1. Comments on the Quality of English Language

The authors have used 'neat' for the control bioceramic materials that are not hybrid. I encourage the authors to use the term 'bioceramics' OR choose another word instead of 'neat.' It is not a scientific word.

- The word “neat” was deleted when used for “pure” bioceramic material according to the reviewer´s suggestion. However, “neat” is kept when is used to distinguish the pure hybrid material from the nBG-loaded composite hybrid. “Neat” is commonly used in scientific literature to distinguish polymers and other pure material matrices from those loaded with particles (composites).

Neat Polymer - an overview | ScienceDirect Topics (T.L. Landsman, A.C. Weems, S.M. Hasan, R.S. Thompson, T.S. Wilson, D.J. Maitland, 20 - Embolic applications of shape memory polyurethane scaffolds, Editor(s): Stuart L. Cooper, Jianjun Guan, Advances in Polyurethane Biomaterials, Woodhead Publishing, 2016, Pages 561-597, ISBN 9780081006146).

Unal H, Mimaroglu A. Comparison of tribological performance of some neat polymer and polymers composites. Surface Engineering. 2013;29(6):455-461. doi:10.1179/1743294413Y.0000000141

Watt, E., Abdelwahab, M.A., Snowdon, M.R. et al. Hybrid biocomposites from polypropylene, sustainable biocarbon and graphene nanoplatelets. Sci Rep 10, 10714 (2020). https://doi.org/10.1038/s41598-020-66855-4

  1. Other than that, minor English editing will polish the manuscript.

- English of the manuscript was proofread and checked.

Reviewer 2 Report

Comments and Suggestions for Authors

I have reviewed the manuscript  Novel organic - inorganic nanocomposite hybrids based on bioactive glass nanoparticles and their enhanced osteoinductive  properties submitted for consideration for publication in Biomolecules. The manuscript is related to the study of the effect of BG on osteoconductive properties of hybrid materials based on SiO2-PTHF  obtained by sol-gel process and polymerization; the paper is easy to read but needs more rigorous analysis of the results and resolving some questions to improve it to be considered for publication in Biomolecules.

Some questions and comments are:

1.      Typographical, grammatical, and spelling errors need to be checked and corrected throughout the manuscript.

Abstract

2.       The abstract should be rewritten concisely and clearly. The abstract should briefly state the purpose of the research, the principal results, and major conclusions.

3.       The abstract should present precise and hard data and avoid misimpressions, i.e: “Hybrids loaded with certain contents of nBG retained the mechanical flexibility ….”

“The results demonstrate that nBG is an effective osteoinductive substance to produce class II hybrid materials….”…Authors must indicate hard data.

Introduction

4.       The introduction must be improved because it should provide a preamble to the topic to be discussed and summarize and highlight the main points to be discussed. How does this research contribute to the state of the art? This should be made clear in the introduction section. Authors must be more specific in the main objective of their work.

5.      There is a lack of adequate research background. The sentence: ……“However, the use of nBGs as nanometric agents to modify the bioactive properties of hybrid materials has not yet been reported”……is an erroneous sentence since there are many investigations in the literature about nBG/hybrids.

6.       The sentence: ….” The composition of the nanocomposite hybrids is optimized as a function of their mechanical flexibility, ability to form bone-like apatite in SBF, cytocompatibility, and capacity to differentiate stem cells toward osteogenic lineage.”…. requires to be reviewed and corrected if necessary. As can be seen from the results section, materials were selected from mechanical flexibility (it must be justified and referenced in the text) and then, authors show results of the ability to form bone-like apatite in SBF, cytocompatibility, and capacity to differentiate stem cells toward osteogenic lineage of that materials. No references about the mechanical flexibility of hybrids nor nBG are presented to justify that.

Materials and methods

7.      Because nBG particles are a fundamental part of the study of this work, the authors must generally mention the obtaining of nBG in the experimental part as well as present relevant results of these for the understanding of the present study.

8.      What is the proportion in weight % of SiO2 in each SiO2-PTHF hybrid? Report that in Table 1.

9.       Are the amounts of 5, 10, and 15% by weight of BG considered concerning 100% of the hybrid? Clarify in the text.

Results and Discussions

10.   A more in-depth analysis of the effect of the nBG content at each SiO2 concentration (TEOS) and the effect of the SiO2 concentration (TEOS) at each nBG concentration is missing for a better understanding of this work.

11.   Based on the results of Figure 1 (strain to failure, compressive strength, and elastic modulus values from all hybrids)  the authors should argue the selection of hybrid materials (0.25TEOS, 5nBG/0.5TEOS, 10nBG/1TEOS, and 15nBG/0.5TEOS) for the study of the ability to form similar apatite to bone in SBF, cytocompatibility and ability to differentiate stem cells towards osteogenic lineage.

12.  Based on the above questions (questions 10 and 11). Is difficult to understand the effect of the amount of nBG on the properties of hybrid materials since they have different compositions to compare them. At least SBF degradability and bioactivity results for all hybrid samples should be presented to determine the best compositions, as these are considered tissue scaffolds and only temporarily remain in the body while the tissue regenerates, and in conjunction with mechanical property values, determine whether the selected samples are indeed appropriate.

13.  No FT-IR bands of vitreous SiO2 (TEOS) are well observed in the hybrid spectra of Figure 2. Since important bands of the IR spectrum overlap, it is important to put the neat PTHF spectrum and the neat SiO2 (from TEOS) spectrum in Figure 2. FTIR spectra require further analysis.

14.  As the author commented, the nanoparticles of BG have a size of ~ 70 nm; please provide evidence of that (particle size distribution). 

15.  Figure 4. To know the presence and a homogeneous particle size distribution of nBG in the hybrid matrix, please provide the EDX mapping of Ca and P.

16.   If the weight loss of the hybrids is due to the degradation products of nBG particles as the authors commented, why are the weight loss of the hybrids of the present work compared to TEOS/PCL and TEOS/PCL/PTHF hybrids? or do they present the same degradation products? justify.

17.  To prove the apatite precipitation on hybrid, is necessary to present XRD since the Ca/P of hydroxyapatite must be  ~1.64 and it is not well observed in the SEM and EDX analysis. No characteristic apatite morphology is observed.

18.   Show through chemical equations how the chemical structure is formed for a better understanding of Figure 11 (propose the reaction mechanism).

Conclusions

19.   Conclusions should be more concrete, concise and highlight the main research findings related to the aim proposed in the manuscript.

20.   The sentence “…Also, different polymer-silica combinations can be explored to produce nanocomposite hybrids with tuned osteogenic properties….”  This phrase is ambiguous since in the present work not all the results related to the osteogenic properties were explored with the different polymer-silica (nor the amount of nBG particles) mentioned in the experimental section and as a consequence, this does not allow a conclusion regarding the composition effect on the ability to form bone-like apatite in SBF, cytocompatibility, and capacity to differentiate stem cells toward osteogenic lineage.

Comments on the Quality of English Language

 Typographical, grammatical, and spelling errors need to be checked and corrected throughout the manuscript.

Author Response

We thank reviewers for their valuable comments, observations, and corrections to improve the quality of our manuscript. In the elaboration of the revised version of the manuscript, all the observations made by the reviewers were carefully considered. The changes made in the manuscript text appear highlighted in colored letters to better clarity. In addition, replies to each reviewer’s comments are indicated below.

I have reviewed the manuscript  Novel organic - inorganic nanocomposite hybrids based on bioactive glass nanoparticles and their enhanced osteoinductive  properties submitted for consideration for publication in Biomolecules. The manuscript is related to the study of the effect of BG on osteoconductive properties of hybrid materials based on SiO2-PTHF  obtained by sol-gel process and polymerization; the paper is easy to read but needs more rigorous analysis of the results and resolving some questions to improve it to be considered for publication in Biomolecules.

Some questions and comments are:

  1. Typographical, grammatical, and spelling errors need to be checked and corrected throughout

- English of the manuscript was proofread and checked.

Abstract

  1. The abstract should be rewritten concisely and clearly. The abstract should briefly state the purpose of the research, the principal results, and major conclusions.

- Abstract was rewritten according to the reviewer’s suggestion.

  1. The abstract should present precise and hard data and avoid misimpressions, i.e:“Hybrids loaded with certain contents of nBG retained the mechanical flexibility ….”

- The precise contents of nBG of the flexible hybrid materials are now indicated.

  1. The results demonstrate that nBG is an effective osteoinductive substance to produce class II hybrid materials….”…Authors must indicate hard data.

- Osteoinductive properties exhibited by the hybrids are now more clearly supported in the abstract by indicating the results of the in vitro osteogenic assays.

 Introduction

  1. The introduction must be improved because it should provide a preamble to the topic to be discussed and summarize and highlight the main points to be discussed. How does this research contribute to the state of the art? This should be made clear in the introduction section. Authors must be more specific in the main objective of their work.

- Introduction was rewritten according to the reviewer’s suggestion.

  1. There is a lack of adequate research background. The sentence: ……“However, the use of nBGs as nanometric agents to modify the bioactive properties of hybrid materials has not yet been reported”……is an erroneous sentence since there are many investigations in the literature about nBG/hybrids.

- To our best knowledge our current work is the first study about the elaboration of a class II organic/inorganic (O/I) hybrid modified with bioactive glass nanoparticles (nBG). The literature review shows that there are many investigations about polymer nanocomposites modified with nBG, including several studies reported by our own group (e.g. Ref [29], [32], [88], [89]) but no using class II hybrids. Nanocomposites are composed of two distinct phases with weak intermolecular interactions between the organic and inorganic components (e.g. nBG particles embedded into a pure polymer matrix). In our current work, nBG has been incorporated into a class II hybrid composed of a single phase formed by PTHF organic chains covalently linked to an inorganic bioactive glass network. Thus, our work constitutes the first effort to improve the osteostimulative properties of class II hybrids by using nanosized BG particles.

6. function of their mechanical flexibility, ability to form bone-like apatite in SBF, cytocompatibility, and capacity to differentiate stem cells toward osteogenic lineage.”…. requires to be reviewed and corrected if necessary. As can be seen from the results section, materials were selected from mechanical flexibility (it must be justified and referenced in the text) and then, authors show results of the ability to form bone-like apatite in SBF, cytocompatibility, and capacity to differentiate stem cells toward osteogenic lineage of that materials. No references about the mechanical flexibility of hybrids nor nBG are presented to justify that.

- The previous optimization of the hybrid composition as function of their mechanical flexibility is now justified and supported with references in the introduction.

Materials and methods

  1. Because nBG particles are a fundamental part of the study of this work, the authors must generally mention the obtaining of nBG in the experimental part as well as present relevant results of these for the understanding of the present study.

- Section “2.3.1. nBG characterization” was added containing SEM (morphology and particle size)/EDX, DLS, XRD, and FTIR analysis of the synthesized nBG particles. Fig. 2 and 4 of the first version already showed the characterization of nBG by FTIR, SEM, EDX, and XRD.

8. What is the proportion in weight % of SiO2in each SiO2-PTHF hybrid? Report that in Table 1.

- Table 1 shows the compositions of synthesis mixture. According to the suggestion of the reviewer, we estimated the nominal weight of SiO2 in the reacting mixture, however SiO2 wt. % is around 0.1 % (because of the high THF quantities used in the reaction). So, probably this information could not be significant for the paper readers. On the other hand, perhaps the reviewer was interested in knowing the final proportions of SiO2/PTHF in the synthesized material, however, these contents (silica-polymer) are not trivial to measure by using conventional analysis techniques. So, we have decided to keep the information shown in Table 1, in addition, EDX compositional analysis of the elements present in the hybrids shown in Fig. 5 contribute to observe the increasing of silica contents as the concentration of nBG increase in the nanocomposite.

  1. Are the amounts of 5, 10, and 15% by weight of BG considered concerning 100% of the hybrid? Clarify in the text.

- The contents of nBG (wt.%) were estimated from total mass of all the reaction constitutes. This is now clarified in the text of the section 2.2.

Results and Discussions

  1. A more in-depth analysis of the effect of the nBG content at each SiO2concentration (TEOS) and the effect of the SiO2 concentration (TEOS) at each nBG concentration is missing for a better understanding of this work.

- The effect of nBG content and TEOS concentration (SiO2 content) is addressed more deeply (Section 3.3, discussion Fig. 2A).

  1. Based on the results of Figure 1 (strain to failure, compressive strength, and elastic modulus values from all hybrids) the authors should argue the selection of hybrid materials (0.25TEOS, 5nBG/0.5TEOS, 10nBG/1TEOS, and 15nBG/0.5TEOS) for the study of the ability to form similar apatite to bone in SBF, cytocompatibility and ability to differentiate stem cells towards osteogenic lineage.

- Since 0.25TEOS, 5nBG/0.5TEOS, 10nBG/1TEOS, and 15nBG/0.5TEOS were the only hybrids that exhibited mechanical flexibility, these materials were selected for structural characterization and assessing their bioactive properties. This justification was now added to the end of section 3.1.

  1. Based on the above questions (questions 10 and 11). Is difficult to understand the effect of the amount of nBG on the properties of hybrid materials since they have different compositions to compare them. At least SBF degradability and bioactivity results for all hybrid samples should be presented to determine the best compositions, as these are considered tissue scaffolds and only temporarily remain in the body while the tissue regenerates, and in conjunction with mechanical property values, determine whether the selected samples are indeed appropriate.

  • We agree with the comments of the reviewer regarding would be easier to observer the nBG effect if the selected nanocomposites were made using the same hybrid matrix. However, as already was argued, the bioactive properties were studied only for those materials that were mechanically flexible, because flexibility is the initial motivation to explore bioactive glass – based hybrids. Even so, within the selected nanocomposites, 5nBG/0.5TEOS and 15nBG/0.5TEOS are made of the same hybrid matrix, so, the effect of increase three times the content of nBG can be analyzed by comparing these materials. Discussions referred to the effect of the content of nBG can be specially found in the discussion of Fig. 9 (SBF), Fig. 11 (ALP activity), and Fig. 12 (Runx2 and osterix).

  1. No FT-IR bands of vitreous SiO2(TEOS) are well observed in the hybrid spectra of Figure 2. Since important bands of the IR spectrum overlap, it is important to put the neat PTHF spectrum and the neat SiO2 (from TEOS) spectrum in Figure 2. FTIR spectra require further analysis.

  • FTIR bands of vitreous SiO2of the hybrid are mentioned in the discussion, and it is indicated that are not easily distinguished from the vibrations of nBG and even overlap with bands of PTHF chains. This discussion has now been improved.

  1. As the author commented, the nanoparticles of BG have a size of ~ 70 nm; please provide evidence of that (particle size distribution). 

- New Fig. 1 was added showing information of nBG characterization, including SEM imagen and particle distribution obtained from both SEM measurements and DLS analysis.

  1. Figure 4. To know the presence and a homogeneous particle size distribution of nBG in the hybrid matrix, please provide the EDX mapping of Ca and P.

- EDX mapping of Ca and P was added in new Fig. 5. However, Si mapping seem to be more sensitive to the distribution of the nBG into the hybrid matrix.

  1. If the weight loss of the hybrids is due to the degradation products of nBG particles as the authors commented, why are the weight loss of the hybrids of the present work compared to TEOS/PCL and TEOS/PCL/PTHF hybrids? or do they present the same degradation products? justify.

- Thank you for the observation. We have now clarified that degradation is produced by both SiO2 and nBG inorganic phases. In addition, it has been clarified that the idea was to emphasize that the degradability of SiO2/PTHF (PTHF is a stable polymeric phase) loaded with nBG is comparable with those of hybrids prepared with more degradable polymers (e.g. PLC), which is consequence of the incorporation of nBG.

  1. To prove the apatite precipitation on hybrid, is necessary to present XRD since the Ca/P of hydroxyapatite must be  ~1.64 and it is not well observed in the SEM and EDX analysis. No characteristic apatite morphology is observed.

- The formation of apatite in SBF was analyzed by FTIR and SEM/EDX, in addition, XRD analysis is included in this revised version. XRD reflections of apatite were not detected, which can be due to a low concentration of the formed mineral phase and/or discrete formation of apatite deposits on the hybrid surface. However, apatite formation was detected by the presence of characteristic FTIR vibrations in all the samples. FTIR is a well-validated surface analysis to detect the crystallization of apatite layer on the surface of bioactive glasses (L. L. Hench, “Characterization of Bioceramics” in “An Introduction to Bioceramics,” (L. L. Hench, Editor), 2nd Edition, Chapter 37, pp. 530-532, Imperial College Press, London, 2013; International standard ISO23317: 2014(E) Implants for surgery- In vitro evaluation for apatite forming ability of implant material). On the other hand, mineralized surfaces exhibit typical morphology of apatite formed from a bioactive glass surface.

EDX is a compositional technique that does not always allow to determine the exact Ca/P ratio of the crystallized apatite, especially when a thin layer of apatite is crystallized on a larger mass of a polymer-based material.

  1. Show through chemical equations how the chemical structure is formed for a better understanding of Figure 11 (propose the reaction mechanism).

-  New Fig. 13 including the chemical reactions involved in the hybrid synthesis is now shown. The discussion was modified accordingly.

Conclusions

  1. Conclusions should be more concrete, concise and highlight the main research findings related to the aim proposed in the manuscript.

- The conclusions were rewritten in a more concise manner according to the reviewer’s suggestion.

  1. The sentence “…Also, different polymer-silica combinations can be explored to produce nanocomposite hybrids with tuned osteogenic properties….”This phrase is ambiguous since in the present work not all the results related to the osteogenic properties were explored with the different polymer-silica (nor the amount of nBG particles) mentioned in the experimental section and as a consequence, this does not allow a conclusion regarding the composition effect on the ability to form bone-like apatite in SBF, cytocompatibility, and capacity to differentiate stem cells toward osteogenic lineage.

- The results presented in this work demonstrate that a hybrid matrix (such as PTHF-SiO2) loaded with nBG has higher osteoinductivity than that of the neat hybrid matrix. Particularly, it is evidenced through the measurements of cell osteogenic differentiation biomarkers that the incorporation of nBG increases the biological ability of the material to induce the process of stem cells toward osteogenic lineage (please compare 0.5TEOS matrix loaded with 5 and 15 %wt. of nBG). Based on these results, the intention of the indicated sentence is suggesting the possibility of exploring other hybrid matrices (different of PTHF-SiO2) in the preparation of n-BG – based nanocomposite hybrids.

 Comments on the Quality of English Language.

Typographical, grammatical, and spelling errors need to be checked and corrected throughout the manuscript.

- English of the manuscript was proofread and checked.

Round 2

Reviewer 2 Report

Comments and Suggestions for Authors

The authors improved the manuscript and almost answer the questions. I have no further comments or suggestions to contribute. Hence, I recommend considering the current manuscript for publication in Biomolecules